# Back to the Future: Immune Protection or Enhancement of Future Coronaviruses

**DOI:** 10.3390/microorganisms12030617

**Published:** 2024-03-19

**Authors:** Merit Bartels, Eric Sala Solé, Lotte M. Sauerschnig, Ger T. Rijkers

**Affiliations:** Science and Engineering Department, University College Roosevelt, 4331 CB Middelburg, The Netherlands; m.bartels@ucr.nl (M.B.); e.salasole@ucr.nl (E.S.S.); l.sauerschnig@ucr.nl (L.M.S.)

**Keywords:** SARS-CoV-2, COVID-19, human coronaviruses, conserved T-cell epitopes, antibody-dependent enhancement, original antigenic sin

## Abstract

Before the emergence of SARS-CoV-1, MERS-CoV, and most recently, SARS-CoV-2, four other coronaviruses (the alpha coronaviruses NL63 and 229E and the beta coronaviruses OC43 and HKU1) had already been circulating in the human population. These circulating coronaviruses all cause mild respiratory illness during the winter seasons, and most people are already infected in early life. Could antibodies and/or T cells, especially against the beta coronaviruses, have offered some form of protection against (severe) COVID-19 caused by infection with SARS-CoV-2? Related is the question of whether survivors of SARS-CoV-1 or MERS-CoV would be relatively protected against SARS-CoV-2. More importantly, would humoral and cellular immunological memory generated during the SARS-CoV-2 pandemic, either by infection or vaccination, offer protection against future coronaviruses? Or rather than protection, could antibody-dependent enhancement have taken place, a mechanism by which circulating corona antibodies enhance the severity of COVID-19? Another related phenomenon, the original antigenic sin, would also predict that the effectiveness of the immune response to future coronaviruses would be impaired because of the reactivation of memory against irrelevant epitopes. The currently available evidence indicates that latter scenarios are highly unlikely and that especially cytotoxic memory T cells directed against conserved epitopes of human coronaviruses could at least offer partial protection against future coronaviruses.

## 1. Introduction

The COVID-19 pandemic outbreak has brought attention to a critical global public health issue that has affected billions of people. The virus that causes COVID-19 is SARS-CoV-2, which displays a high degree of similarity to the SARS-CoV species that caused the 2003 coronavirus outbreak. This has greatly aided the identification and full genetic characterization of SARS-CoV-2 [1]. The rapid spread of the illness necessitated the development of preventative and therapeutic measures for the care of SARS-CoV-2-infected individuals. Thanks to the development and implementation of effective vaccines, and probably also the emergence of the omicron variant, the pandemic could be brought under control. On 17 March 2023, the Director-General of the WHO indicated that he was confident that in 2023, COVID-19 would be over as a public health emergency of international concern [2] and, indeed, in May 2023, the WHO declared that COVID-19 is now an established and ongoing health issue which no longer constitutes a public health emergency of international concern [3]. SARS-CoV-2, however, has not disappeared from the world and still causes severe disease in the most vulnerable. In an optimistic scenario, it can be envisioned that within time, SARS-CoV-2 can be added to the so-called cold cousins, i.e., the circulating coronaviruses that cause the common cold during winter seasons (229E, OC43, NL63, and HKU1 [4]). The human population then would be relatively protected against future variants of SARS-CoV-2. It has to be emphasized that the timeline of the evolutionary conversion of a severe acute respiratory coronavirus to a common cold coronavirus is difficult to predict and could potentially take tens to hundreds of years. In a pessimistic scenario where novel coronaviruses adapt to humans, immunological memory would not protect against novel coronaviruses but instead, due to antibody-dependent and related mechanisms, could enhance the severity of the disease. In this paper, currently available evidence will be discussed, which suggests that T-cell memory, directed against conserved epitopes on the spike protein and nucleocapsid protein, could protect against future coronaviruses.

Coronaviruses (CoVs) are positive-sense, single-stranded RNA viruses; they are members of the family Coronaviridae, subfamily Coronavirinae, and order Nidovirales. CoVs possess one of the largest genomes among all RNA viruses. Four species of coronaviruses, including alpha, beta, gamma, and delta, can be distinguished. SARS-CoV-2 belongs to the species severe acute respiratory syndrome-related coronavirus [5]. Originally, they were categorized based on serology; however, the categories named above are based on phylogenetic clustering [6].

The spike, envelope, membrane, and nucleocapsid are the four main structural proteins that make up the coronavirus virion structure (Figure 1). Although the number and location of accessory ORFs present in different coronavirus species vary, distinct coronavirus strains share a common genetic organization for the coding region, encoding for a canonical set of genes in the order 5′ end–ORF1a/b replicase, spike, envelope, membrane, nucleocapsid–3′ end. The subgenomic mRNAs that carry out gene translation combine with the viral genome to form a 5′ and 3′ co-terminal nested set. Subgenomic mRNAs are present along with a common 5′ leader sequence and a 3′ terminal sequence. The genome contains small untranslated regions at both the 3′ and 5′ ends. In addition, the viral genome encodes a number of nonstructural proteins (NSPs), such as the papain-like protease (PLpro), the coronavirus main protease (3CLpro), and the RNA-dependent RNA polymerase (RdRp) [7].

Before the emergence of SARS-CoV-1, Middle Eastern respiratory syndrome coronavirus (MERS-CoV), and most recently, SARS-CoV-2, four other coronaviruses had already been circulating in the human population: the alpha coronaviruses NL63 and 229E and the beta coronaviruses OC34 and HKU1 (SARS-CoV-1, MERS-CoV, and SARS-CoV-2 are also beta coronaviruses) [8]. The circulating coronaviruses all cause mild respiratory illness during the winter seasons (which is the reason why they are sometimes called the “cold cousins” of SARS-CoV-2), and most people are already infected in early life [9,10]. Could antibodies or T cells (especially against the beta coronaviruses), which have originally been generated after exposure to a circulating coronavirus, have offered some form of protection against (severe) COVID-19 caused by SARS-CoV-2 infection? Related is the question of whether survivors of SARS-CoV-1 or MERS-CoV would be relatively protected against SARS-CoV-2.

## 2. The Immune Response to SARS-CoV-2 Infection and Vaccination

Upon infection with SARS-CoV-2, an innate immune response is initiated in the respiratory tract. This involves massive activation of mainly proinflammatory cytokines [11,12]. The size of the initial viral load and the effectiveness of the innate immune response—especially the type I interferon-mediated response—appears to be crucial in determining the course of the subsequent adaptive response and the final clinical outcome [13]. Effective interferon signaling plays a critical role in acute infection, as demonstrated by both acquired and genetic factors. The indicators of severe clinical outcomes include early and persistent inflammation with elevated interferon (IFN)-α, TNF, and IFN-γ, as well as a slow decline in the viral load [14].

Like other viral respiratory infections, SARS-CoV-2 infection promotes the fast production of IgM, IgG, and IgA antibodies. These antibodies, including those that bind to the spike protein and nucleocapsid, are detectable in the sera as early as one-week post-infection. The speed at which these reactions occur suggests that the antibodies originate from extrafollicular differentiation of naive B cells into short-lived antibody-secreting cells, independent of the traditional germinal center reaction [15]. These antibodies also show neutralizing activity against live or pseudotyped SARS-CoV-2; the latter activity can be readily detected in convalescent sera, though the degree of neutralization that can be achieved varies widely between people. The inconsistent outcomes of plasma therapy, which was tried early in the pandemic, could be partially explained by this variability [16].

### 2.1. The Humoral Immune Response to SARS-CoV-2 Infection

Hashem et al. studied the relationship between the severity of disease and the antibody titers in COVID-19 patients [17]. It was found that COVID-19 patients who experienced severe or moderate infections exhibited noticeably higher total neutralizing antibody (nAbs) levels. Furthermore, compared to mild cases, severe cases had significantly higher levels of IgG antibodies directed against the S1 subunit of the spike protein (S1). Notably, anti-N IgG and IgM levels were induced at higher levels in moderate and severe cases compared to mild infections, and these levels were significantly correlated with the severity of the disease [18]. Anti-S1 and N antibodies (IgG and IgM), as well as nAbs, were found to be significantly higher in patients who required ICU admission or had fatal outcomes when patients were stratified based on their need for ICU admission or infection outcome. Similar findings were also published by Liu et al. and Rijkers et al., who reported that nAbs were higher in hospitalized COVID-19 patients as compared to healthcare workers with mild clinical symptoms, not requiring hospitalization [19,20]. These findings unequivocally demonstrate the relationship between the severity of the COVID-19 infection and the activation state of the humoral immune system as a whole.

Neutralizing antibodies are mostly directed against the receptor-binding domain (RBD) of the SARS-CoV-2 spike protein (indicated in Figure 2) but can also be directed against the N-terminal domain or the stem helix region or the fusion peptide region in the S2 subunit of the spike protein [21]. Evolutionary variability of the virus, reflected in the successive appearance of alpha, delta, and omicron variants, caused changes in an increasing number of amino acids within the RBD and a concomitant loss of the neutralizing capacity of existing antibodies.

#### COVID-19 and Response to Vaccination in Patients with Humoral Immunodeficiency or B-Cell-Depletion Therapy

The importance of the role played by humoral immunity in COVID-19, as well as a correlation of protection against COVID-19 after vaccination, remains uncertain because SARS-CoV-2 infection or vaccination induces both a humoral and cellular immune response. Patients with agammaglobulinemia lack (functional) B cells but have an intact cellular immune system. The immune response of agammaglobulinemia patients during SARS-CoV-2 infection and the outcome of COVID-19 could serve as a model to further investigate the relative role of humoral and cellular immunity.

Soresina et al. described the cases of two X-linked agammaglobulinemia (XLA) patients who developed pneumonia as a clinical manifestation of SARS-CoV-2 infection but did recover without requiring oxygen ventilation or intensive care treatment [25]. The cases of two adolescent male XLA patients were discussed by Devassikutty et al., where, except for delayed recovery, both patients had successful outcomes [26]. In a paper on seven patients (two with agammaglobulinemia and five with common variable immunodeficiency (CVID)), it was noted that a milder course of COVID-19 was observed in agammaglobulinemia as compared to the CVID patients. Furthermore, their COVID-19 course had a shorter duration and required no need for anti-inflammatory treatment with an IL-6-blocking drug [27]. These data, on an admittedly low number of patients, indicate that in the complete absence of B cells, such as in the case of XLA, COVID-19 does not necessarily take a severe course. It could be argued that XLA patients are substituted with intravenous immunoglobulins (IVIG) and that antibodies contained within the preparation confer protection against COVID-19. The above-mentioned studies, however, were all published in the first year of the pandemic, and at that time, IVIG preparations did not (yet) contain SARS-CoV-2 IgG antibodies.

From these findings in patients with B-cell deficiencies, it can be speculated that B-cell depletion may not have a major detrimental effect on COVID-19 recovery. In B-cell lymphoma patients treated with rituximab and bendamustine, the humoral immune response (but not the T-cell response) to SARS-CoV-2 vaccination is impaired, as would be expected from a B-cell-depleting treatment [28]. Similar results were published by Candon et al. and Ishio et al. [29,30]. Patients with B-cell lymphoma who were successfully treated with rituximab and recovered show normal antibody responses after (mRNA) SARS-CoV-2 vaccines [31]. Finally, in a group of patients with childhood-onset nephrotic syndrome who received rituximab for its steroid/calcineurin-inhibitor sparing effect, SARS-CoV-2 antibodies largely persisted. This indicates that long-lived plasma cells play a major role in maintaining antibody levels [32].

Most of the patients with multiple sclerosis who were treated with rituximab had a mild course of COVID-19 [33]. In a French cohort study on COVID-19 in patients with inflammatory rheumatic and musculoskeletal diseases treated with rituximab, severe disease occurred more frequently than in non-rituximab-treated patients [34]. For other rituximab indications, variable effects on the incidence and severity of COVID-19 have been reported, but an extensive discussion of these data is beyond the scope of this paper.

### 2.2. T-Cell-Mediated Immunity against Coronaviruses

Probably fueled by the massive reports in the popular press on vaccination and immunity, the general conception is that mainly antibodies directed against the SARS-CoV-2 spike protein offer protection against disease. Furthermore, antibody levels should remain high in order to stay protected. The role of T-cell-mediated immunity against SARS-CoV-2, and viruses in general, has received relatively little attention. Of the papers dealing with SARS-CoV-2 immunity, 87% concern humoral immunity, and only 13% address T-cell immunity.

#### Importance of T-Cell Immunity

T cells, in principle, can respond to any viral peptides, including those of more conserved regions. SARS-CoV-2, as well as SARS-CoV-1, for that matter, can enter the host cells through binding to the ACE2 receptor. Upon entry, viral capsid proteins are broken down by enzymatic degradation, setting free the genetic material and exploiting the host cellular machinery for viral replication purposes [35]. Simultaneously, viral proteins are further processed by the proteasome, after which any peptide that fits can be bound in the groove of an MHC class I molecule (HLA-A, -B, -C) and, upon cell surface expression, be presented to CD8^+^ T cells or bind to MHC class II (HLA-DR, -DP, DQ) [36] molecules on antigen-presenting cells (APCs) to CD4^+^ T-helper cells. The selective interaction with the TCR elicits either a cytotoxic response by CD8^+^ T cells or the activation of CD4^+^ T-helper cells that are necessary for proper stimulation of B-lymphocytes and antibody production [37]. Because all viral peptides, including those derived from non-structural regions, have the potential to elicit a T-cell response, they play a determining role in adaptive cellular immunity. Tarke et al., by using overlapping peptides spanning all structural and non-structural viral proteins, identified several hundred T-cell epitopes for CD4^+^ T cells and CD8^+^ T cells, highlighting the diverse T-cell response to the virus [38]. Additionally, abortive infections can exist where T-cell responses may clear the virus before sufficient viral replication or antibody production has taken place. This process is explored in a study by Swadling et al., who found that the T cells involved in this process are mainly directed against non-structural proteins of the replication-transcription complex [22]. It should be kept in mind that most studies on T-cell epitope mapping are performed with peripheral blood T cells. The T cells that have infiltrated infected tissue, such as the lung, may have a different specificity pattern [39].

The receptor-binding domain of the spike protein of SARS-CoV-2 is localized in the N-terminal region of the protein and contains 17 amino acids that directly interact with the ACE2 receptor in human host cells (Figure 2). Within the conserved regions of the spike proteins of the other hCoVs, there is little sequence homology. Homologies are more prominent in the C-terminal regions of the spike proteins, which is also the part where most CD8^+^ T-cell epitopes are found (Figure 2) [40].

As indicated above, the spike protein contains many T-cell epitopes, of which three can be considered major epitopes, as defined by Ferretti et al., termed YLQ, KCY, and QYI [40]. All three epitopes are located outside the RBD of SARS-CoV-2. The QYI epitope is highly conserved in SARS-CoV-1 and MERS-CoV, as well as in the other hCoVs. Furthermore, in the variants of SARS-CoV-2, from alpha to omicron, including the omicron subvariants, the QYI epitope remains unchanged. However, the epitope closest to the RBD (KCY) has been mutated in the omicron variants [23].

The nucleocapsid (N) protein is well conserved and has been shown to display a high degree of homology between different coronavirus strains [24]. N proteins are associated with the viral RNA (see Figure 1) and are not accessible for antibodies in an intact virus particle. Therefore, anti-N antibodies, which are produced in large quantities following infection, cannot prevent the spread of the virus within the body. N proteins can, however, serve as important T-cell epitopes [41]. Six major epitopes have been identified in SARS-CoV-2, as indicated in Figure 3. Furthermore, all six epitopes are identical in SARS-CoV-1 and SARS-CoV-2 (including the delta and omicron variants of SARS-CoV-2), plus they are located within conserved regions of the different beta coronavirus strains. More specifically, the ASA epitope is conserved in all known human beta coronaviruses and SPR to a lesser degree (Figure 3). The MEV epitope is identical in SARS-CoV-1 and SARS-CoV-2 but not found in other human beta coronaviruses [42]. SARS-CoV-1 survivors showed a robust T-cell response when activated in vitro with N-protein epitopes, including ASA and SPR [42]. This is evidence supporting the suggestion that memory T-cells from previous infections with circulating coronaviruses could have been reactive to SARS-CoV-2 and thus have aided in the cellular protection against severe disease. Any possible existing cross-reactive T-cell immunity or cross-reactive antibodies against SARS-CoV-2 could therefore have resulted from prior infection with another coronavirus, i.e., SARS-CoV-1 or MERS-CoV [18,43], and this would have had a much bigger impact than one of the common cold viruses, OC43, HKU1, NL63, or 229E10 [10,44,45,46]. In a similar scenario, long-lasting cellular immunity could be provided through cross-reactivity against homologous epitopes for new coronaviruses in the future.

When targeting cellular immunity, the focus should be on MHC class I epitopes, which are present on all nucleated cells [47], including those with ACE2 receptors. Displaying viral peptides via MHC I elicits a direct response of cytotoxic CD8^+^ T cells, leading to programmed cell death of the virally infected cell [48]. This immediately halts viral replication as the host cell is necessary for the production of the virus. In COVID-19, an early cytotoxic T-cell response has been observed to correlate with efficient viral clearance and a mild disease course [14,39,45,46]. In addition, CD8^+^ memory T cells generated by previous infection with coronaviruses facilitate a quick response to reinfection by the same strain and might aid in fighting infection by a new strain or entirely new virus [48], provided there would be sufficient sequence homology between relevant epitopes. Therefore, T-cell immunity and especially MHC class I epitopes should receive careful consideration when designing vaccines in the future [45,49].

Furthermore, Verma et al. found that peptides of the N protein can bind with high affinity to TLR4, expressed on monocytes and macrophages [41]. Triggering of TLR4 could be beneficial as this improves antigen uptake and presentation to both CD4^+^ T cells, stimulating the B-cell response and antibody production as well as to CD8^+^ T cells [50]. Hence, identifying and utilizing epitopes that are recognized by APCs additionally to providing binding sites for MHC I could thus promote both cellular and humoral immunity. As indicated above, apart from the spike and N proteins, many T-cell epitopes in SARS-CoV-2 are also found in non-structural parts (NSPs) like ORF1ab or ORF3a [38,51]. However, immunodominant are epitopes of the N protein, giving rise to the highest frequency of specific T-cells [14,52]. Overall, the indicated (immuno-)dominant HLA alleles in response to SARS-CoV-2 are HLA-A*01:01, HLA-A*02:01, HLA-A*03:01, HLA-A*11:01, HLA-A*26:01, HLA-A*68:01, HLA-B*07:02, HLA-B*08:01, and HLA-B*35:01 [38,40]. These HLA alleles specifically have been shown to interact with a wide variety of different structural and non-structural SARS-CoV-2 peptides, and at least one of these prominent HLA class I alleles is present in about 85% of the world’s population [40]. All SARS-CoV-2 T-cell epitopes discussed in this paper can interact with at least one of these aforementioned alleles.

## 3. Potential Negative Impact of Existing Antibodies and Memory B Cells on Future Viruses

When, as discussed above, a novel emerging coronavirus would have considerable antigenic differences with circulating coronaviruses, existing antibodies and memory B cells would offer no protection. It is even possible that existing antibodies and memory B cells would have a negative effect and cause more severe clinical expression of the infection. Two different but partly overlapping mechanisms could be involved. The first one is antibody-dependent enhancement, and the second is the original antigenic sin.

### 3.1. Antibody-Dependent Enhancement (ADE)

Antibody-dependent enhancement (ADE) refers to the enhancement of disease due to the presence of antibodies to the pathogen that has caused the disease. These antibodies could have been induced by previous exposure to the given pathogen, which caused a primary immune response, or previous vaccination, which has induced the production of antibodies. Also, passively administered antibodies (either prophylactic or therapeutic) could potentially lead to ADE. As far as coronaviruses are concerned, there has been a precedent of ADE after vaccination. Cats vaccinated against feline infectious peritonitis virus, a coronavirus, did produce anti-viral antibodies, but after being challenged with the live virus, they were more susceptible than the non-vaccinated control animals [53,54]. While ADE is of concern to vaccine development in general, it was especially relevant for coronavirus vaccines (see above). Thus, the interest in ADE rose during the SARS-CoV-2 pandemic (Figure 4).

#### 3.1.1. In Vitro Studies on Antibody-Dependent Enhancement

Wan et al. mentioned the mechanism by which previous coronaviruses have undergone ADE, specifically MERS-CoV and SARS-CoV [55]. Using monoclonal antibodies specifically targeted to the receptor-binding domain (RBD) of SARS-CoV, Wan et al. found that entry into CD32A-expressing cells was mediated by this monoclonal antibody; however, it blocked entry into cells that expressed the ACE2 receptor [55]. This inhibitory effect serves as a possible mechanism by which ADE could occur with blood lymphocytes since they show poor ACE2 expression as compared to tissue-resident macrophages.

#### 3.1.2. In Vivo Indications for Antibody-Dependent Enhancement

In vivo evidence of ADE has been observed and studied in dengue virus infections. The proposed mechanism for the ADE of dengue virus entails a primary infection of a specific serotype during which antibodies are produced that neutralize the serotype of the primary infection [56]. ADE then occurs during a secondary dengue infection of a different serotype, and the same antibodies no longer function to the original extent. Instead of neutralizing the viral particles, they potentiate the entry into leukocytes [56]. This is achieved as the Fab part of these antibodies will bind to the viral particles, leaving the Fc part open to interact with Fcγ receptors on cells of the immune system. This will then allow leukocytes to recognize and neutralize the virus in normal function; however, in ADE, the Fc part of the antibody will mediate the entry of the viral particles into leukocytes, particularly the macrophages, which will lead to viral replication inside of the leukocytes and ultimately enhancement of the disease. This mechanism of disease enhancement is similar to that discussed in the case of coronaviruses (see above) [57]. The specific glycosylation states of IgG antibodies generated during a SARS-CoV-2 infection have been linked to how the course of the disease will progress ultimately. Afucosylation of the IgG antibodies leads to an increase in the affinity to FcγRIIIA (CD16a) and, thus, an increase in antibody-dependent cellular cytotoxicity (ADCC) [58,59]. Other modifications to antibodies and different antibody classes need to be researched to elucidate the roles that galactosylation and sialylation have on complement activation and disease progression, which could play a role in ADE through the aforementioned mechanism.

When ADE occurs in cases of SARS-CoV-2 infection, it is expected that after vaccination, COVID-19 will take a more severe course. However, studies on breakthrough infections [60,61] as well as case–control studies [62] have shown that vaccinated individuals with breakthrough COVID-19 infections experience milder symptoms, with fewer hospital and ICU admission and lower mortality rates as compared to unvaccinated individuals [63].

#### 3.1.3. Proposed Mechanisms of Antibody-Dependent Enhancement

Fc gamma receptors (FcγRs) are involved in the specific signaling pathways that purportedly cause ADE and are primarily expressed on the surface of leukocytes. If the binding of the Fc part of IgG would mediate viral entry into the leukocytes, on top of the capacity of the virus to enter target cells expressing ACE2 receptors, this could lead to ADE (Figure 5).

Upon engagement with immune complexes formed by antibodies and their corresponding antigens, FcγRs initiate intracellular signaling cascades, leading to various cellular responses, including phagocytosis: antibody-dependent cellular phagocytosis, ADCP [64], production of pro-inflammatory cytokines and cellular activation [65]. In the context of ADE, these signaling pathways may inadvertently contribute to increased infection by promoting the uptake of virus-containing immune complexes. Therefore, in ADE, the phagocytic cells could serve as a (additional) host for the virus.

Also, the complement system may play a critical role in ADE. There are two main mechanisms involving complement that have been exploited by viruses, namely a C1q-mediated mechanism [66,67] and a C3-mediated mechanism [68,69] (Figure 5c,d).

### 3.2. The Original Antigenic Sin

Apart from ADE, another potential negative effect of immunological memory for future coronaviruses would be the so-called original antigenic sin [70]. This phenomenon was first described by Thomas Francis Jr. in 1960 and states that a new strain of an existing virus (in his case, influenza virus) could activate memory B cells specific for a previous strain. Those antibodies could bind to the new strain but not neutralize the virus [71]. This phenomenon needs to be considered when designing and evaluating current and future coronavirus vaccines [72]. Current data do not point toward a role for “original antigenic sin” in the immune response to SARS-CoV-2 infection or vaccination [73,74].

Within the T-cell system, an equivalent of ADE or original antigenic sin has not been described. Any negative effect of existing T-cell memory on protection against disease caused by future coronavirus infections, therefore, is not to be expected.

## 4. Concluding Remarks and Outlook for the Future

The best model to assess the protective effect of existing immunological memory for future coronaviruses is to compare COVID-19 incidence and severity in survivors of SARS-CoV-1 or MERS-CoV infection. In SARS-CoV-1 survivors, cross-reactive SARS-CoV-2 IgG antibodies can be detected, and these SARS survivors also generate a much stronger antibody response after SARS-CoV-2 vaccination [75]. In a retrospective cohort study, it was found that symptomatic MERS-CoV patients were at a lower risk for COVID-19 [76]. Care must be taken when interpreting these data because SARS-CoV-1 and MERS-CoV patients survived without being vaccinated or specifically treated. Therefore, they can have an immune system that was and remained a priori better equipped to combat coronaviruses.

Our analysis of the literature shows and confirms that cytotoxic T cells against conserved epitopes on the SARS-CoV-2 spike protein and nucleocapsid protein are expanded during infection and do limit the severity of COVID-19 [14,77]. Based on these findings, it is possible that existing memory T cells, generated during exposure to circulating hCoVs, including SARS-CoV-1 and MERS, have offered partial protection against COVID-19 caused by SARS-CoV-2 infection and thus could have contributed to a lower case-fatality rate of SARS-CoV-2 as compared to SARS and MERS. Therefore, it should be considered to include conserved epitopes of both the spike protein and nucleocapsid protein in future vaccines in order to induce a robust and lasting T-cell response against these relevant epitopes [77]. Appelberg et al. have shown that in their prototype universal SARS-CoV-2 vaccine, the inclusion of only the relevant nucleoprotein T-cell epitopes provided 60% protection against lethal infection in mice [78]. Future coronavirus vaccines ideally would be pancoronavirus vaccines [79]. Such a vaccine cannot be solely based on spike proteins or epitopes thereof because not all coronaviruses use ACE2 as a cellular receptor. T-cell epitopes of the nucleocapsid protein are conserved among the current hCoVs, and indeed, those epitopes are included in a number of pancoronavirus vaccines currently under development [80].

Regarding the outlook for the future, it should be noted that this review is restricted to an analysis of the immunological factors that may protect against future coronaviruses or enhance disease severity. Other factors, such as virological, societal, and environmental, can be as important or even more important than immune memory and cross-reactivity as such.

Infection with SARS-CoV-2, or vaccination for that matter, induces a strong response of both the humoral and cellular immune systems. In the assessment of the immune status of patients with COVID-19, as well as in the analysis of the immune response to vaccination, emphasis is mostly placed on the quantitation of (neutralizing) antibodies. Although no protective antibody titers have been established, the magnitude of the antibody response is taken as a correlate of protection. T-cell immunity appears to be more important and is directed to epitopes of viral proteins, which are largely conserved among the hCoVs, including the variants of SARS-CoV-2. It, therefore, would be appropriate to (also) implement T-cell-based diagnostic assays. Development and implementation of these assays will be part of preparing for the future.

In their search for the origin of SARS-CoV-2, Temmam et al. have sampled Rhinolophus bats in northern Laos [81]. Many previously unknown coronaviruses were found, including five sarbecoviruses. Three of those (BANAL-52, BANAL-103, and BANAL-236) are close relatives of SARS-CoV-2 and can bind to human ACE2 [82]. These data show that bats, as well as many other animal species for that matter, are a large reservoir from which new coronaviruses can emerge with the potential to set off a new pandemic.

Thanks to the advancement of medical sciences, the original Darwinian principles of struggle for life and survival of the fittest no longer hold completely true. Yet, despite vaccines, diagnostics, and advanced treatment, the burden of infectious diseases, including SARS-CoV-2, weighs on those with an impaired immune system. The “fitness” of the immune system, however, is not as easy to determine as, for instance, that of the cardiac or respiratory system. Unfortunately, there is no simple litmus test for the immune system.

The research team of Amy Huei-Yi Lee has developed a multiomic data integration tool that has the potential to create a kind of immune signature that could predict response to vaccination [82]. On a much larger scale, an initiative has been taken to collect thousands of immune parameters (cells, proteins, genes) of hundreds of thousands of people across the globe in a project called the Human Immunome Project (https://www.humanimmunomeproject.org/ (accessed on 8 January 2024)) [83]. Advancements like these will make it possible to predict whether or not a new virus, or any other microorganism, would have the potential to give rise to endemic or pandemic outgrowth.

## Figures and Tables

**Figure 1 microorganisms-12-00617-f001:**
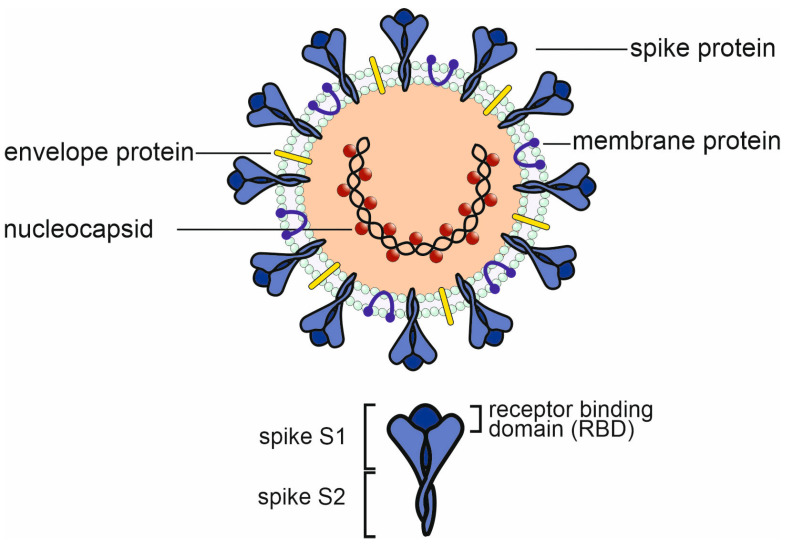
Structure of SARS-CoV-2 with major structural proteins. The immunologically relevant domains of the spike protein (S1, S2, and RBD) are indicated.

**Figure 2 microorganisms-12-00617-f002:**
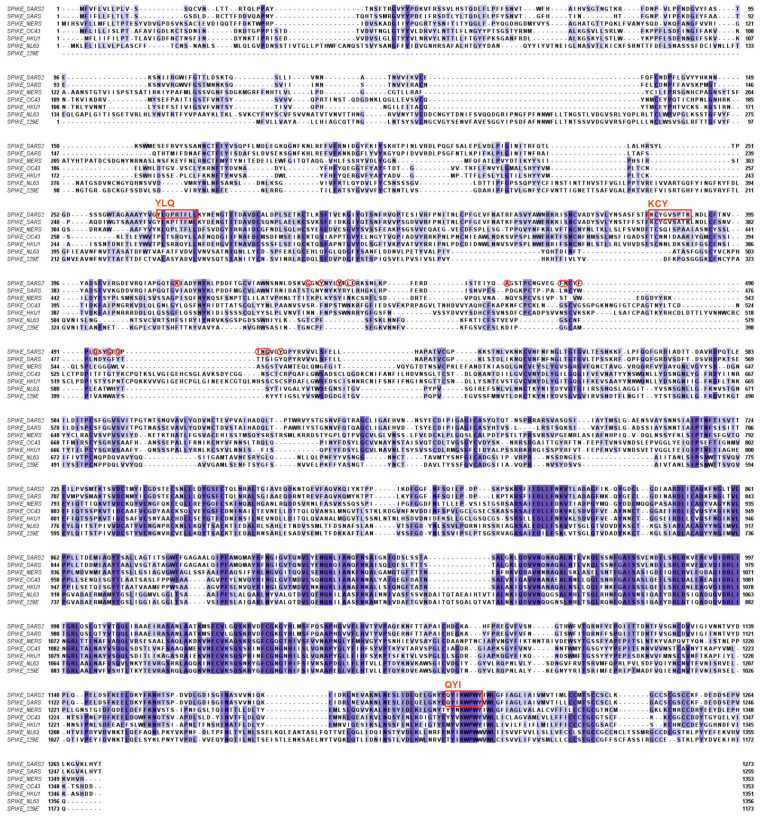
Homology between spike proteins of the human coronaviruses. The amino acids constituting the ACE2-receptor-binding domain of SARS-CoV-2 are encircled in red. Major CD8+ T-cell epitopes are indicated by red rectangles and their first three amino acids according to the SARS-CoV-2 sequence. YLQ = YLQPRTFLL (269–277); KCY = KCYGVSPTKL (378–386); QYI = QYIKWPWYI (1208–1216) [22]. Shades of blue indicate different degrees of identity between the compared protein sequences. Figure 2 and Figure 3 were created using Clustal Omega version 1.2.2 (http://www.clustal.org/omega/ (accessed on 30 January 2024)) and Jalview sequence alignment software version 2.11.2.0 (https://www.jalview.org/ (accessed on 30 January 2024)). All viral peptide sequences were retrieved from the UniProt database (https://www.uniprot.org/ (accessed on 30 January 2024)).

**Figure 3 microorganisms-12-00617-f003:**
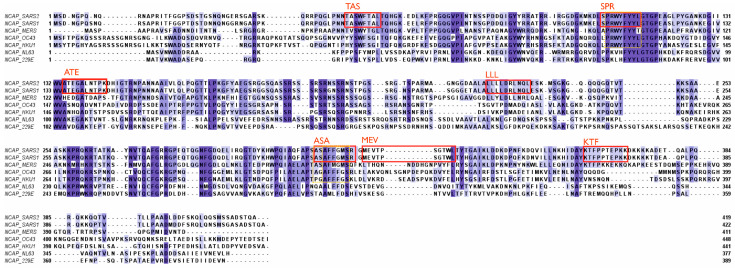
Major T-cell epitopes within the nucleocapsid protein. All epitopes are indicated by their first three letters. TAS = TASWFTAL (49–56); SPR = SPRWYFYYL (105–113); ATE = ATEGALNTPK (134–143); LLL = LLLDRLNQL (222–230); ASA = ASAFFGMSR (311–319); KTF = KTFPPTEPKK (361–369); MEV = MEVTPSGTWL (322–331). Further details are provided in the legend of Figure 2. Based on data from [22,23,24].

**Figure 4 microorganisms-12-00617-f004:**
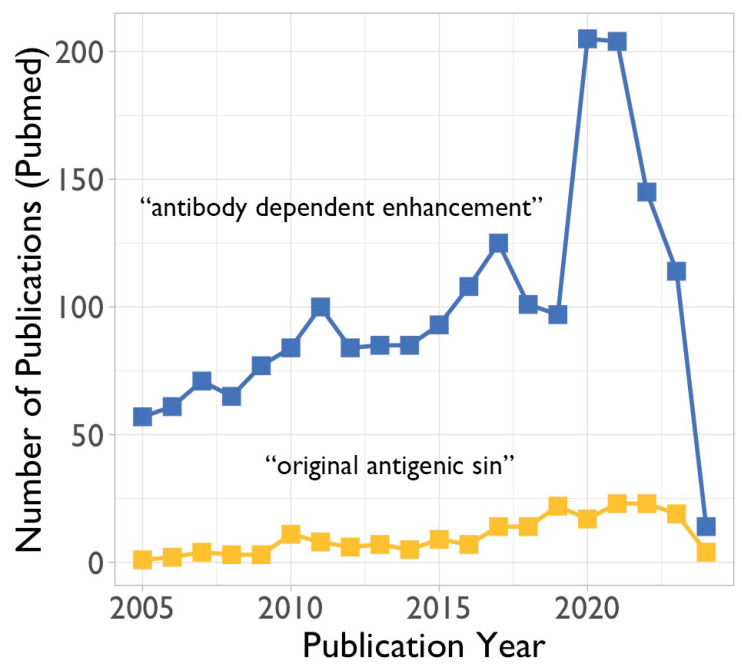
Number of publications per year retrieved from PubMed (https://pubmed.ncbi.nlm.nih.gov/ (accessed on 14 January 2024)) using the search terms “antibody dependent enhancement” or “original antigenic sin” during the past 20 years.

**Figure 5 microorganisms-12-00617-f005:**
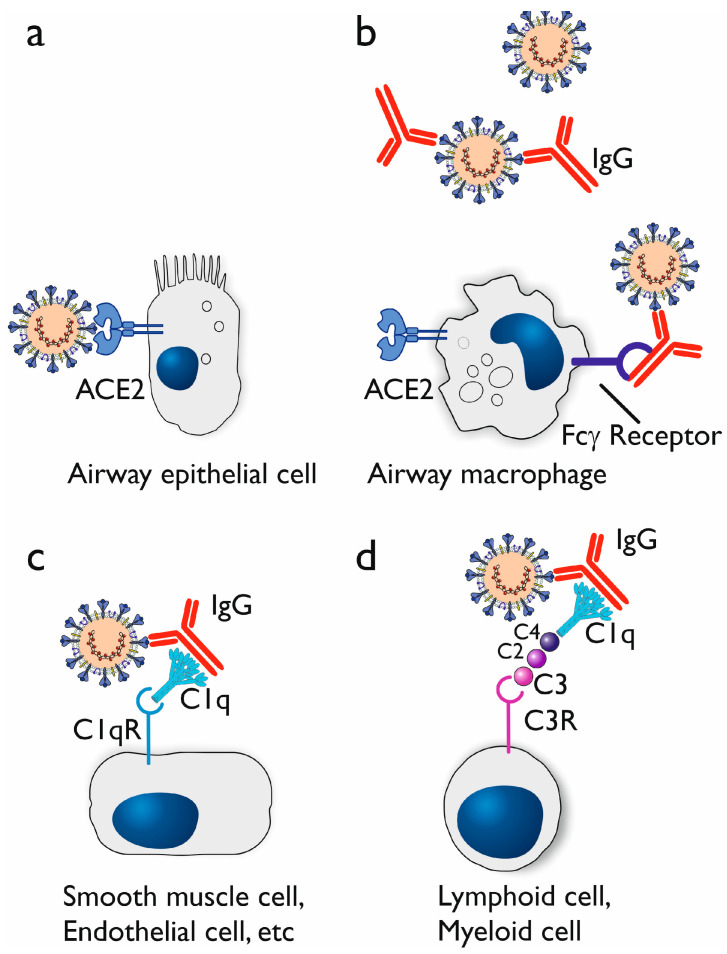
Cellular tropism of SARS-CoV-2 and the potential mechanism of antibody-dependent enhancement. Human cells expressing the angiotensin-converting enzyme 2 (ACE2), such as airway epithelial cells (panel (**a**)) and airway macrophages (panel (**b**)), can be infected by SARS-CoV-2. When IgG antibodies (against viral surface proteins such as the spike protein) are generated, these can form immune complexes with the virus. The Fc part of the IgG can be bound by cellular Fc**γ** receptors, which enables the virus to enter the cell via this route (panel (**b**)). When the complement system is activated, C1q binds to the IgG, and the complex can be captured by C1q receptors (C1qR) (panel (**c**)). Ongoing complement activation leads to binding and activation of C4, C2, and C3 (C1r and C1s are not shown). C3 receptors (C3Rs), which include complement receptor 1 (CR1), CR2, and CR3, can also bind the complex of virus, antibody, and complement proteins (panel (**d**)).

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
