# Peer review of "Back to the Future: Immune Protection or Enhancement of Future Coronaviruses"

_microorganisms, 2024, doi:10.3390/microorganisms12030617_

Round 1
Reviewer 1 Report
Comments and Suggestions for Authors
The authors Bartels et al. ask the question whether previous exposure to endemic coronaviruses or to SARS-CoV-2 virus or vaccine causes protection against future SARS-CoV-2 or other coronavirus infections.
However, the manuscript contains several paragraphs unrelated to these questions.
For example:
What has the relationship between the disease severity and the antibody titers to do with cross-protection?
Why is IVIG or passive vaccination mentioned in this context?
The number of publications regarding a certain topic is irrelevant.
Additionally the relevant publications that had addressed these questions experimentially are amiss.
Just to provide some names of authors that had published ground-breaking data related T/B cell cross-reactivity and have to be mentioned in a related review:
- Mateus/Grifoni/Weiskopf/Sette/Crotty
- Swadling/Maini
- Kundu/Lalvani
- Braun/Loyal/Thiel
- Song/Andrabi
- Ng/Kassiotis
- Niessl/Sekine/Buggert
- LeBert/Bertoletti
Unfortunately, all questions addressed in this review are extensively and more qualitatively discussed elsewhere (especially by authors from the list above).
I recommend the authors to significantly overwork the manuscript and either change the focus or stratify the content.
Comments on the Quality of English Language
-
Author Response
Reviewer #1
Thank you very much for taking the time and effort to review our manuscript. Please find the detailed point by point responses below. We have indicated the positions in the (clean version of the) manuscript where the changes were made. The corresponding revisions/corrections can also be found highlighted/in track changes in the re-submitted files.
Comments and Suggestions for Authors
The authors Bartels et al. ask the question whether previous exposure to endemic coronaviruses or to SARS-CoV-2 virus or vaccine causes protection against future SARS-CoV-2 or other coronavirus infections.
Comment: However, the manuscript contains several paragraphs unrelated to these questions.
Response: We are sorry that we have been unclear about the relevance of a number of paragraphs in our manuscript. In the response to the specific questions and remarks given below, we have explained why those paragraphs were included in the manuscript, and where necessary we have revised the text of the manuscript.
Comment: What has the relationship between the disease severity and the antibody titers to do with cross-protection?
Response: In this paragraph we describe the nature of the immune response to infection with SARS-CoV-2. We feel it is important to emphasize that during active and severe disease, both innate responses, as well as humoral and cellular immunity are activated. While this is general knowledge, it serves as introduction to the subsequent parts on the relative contribution of humoral and cellular immunity to overall protection against severe disease.
Comment: Why is IVIG or passive vaccination mentioned in this context?
Response: IVIG and passive vaccination were included because they serve as models to evaluate the relative contribution of (effector molecules of the) humoral immune system to protection against a viral infection, in this case SARS-CoV-2 (and SARS-CoV-1).
After lengthy discussions, and also because other reviewers had similar comments on inclusion of these sections, we have deleted these paragraphs from the revised manuscript.
Comment: The number of publications regarding a certain topic is irrelevant.
Response: Although we wouldn’t put it as strong as that “the number of publications on a certain topic is irrelevant”, we do agree that a separate Table on the number of papers on SARS-CoV-2 antibodies and T-cells is kind of redundant. We therefore have removed this Table from the revised manuscript and edited the corresponding text. For “antibody dependent enhancement” we feel differently. This phenomenon received quite some attention in the lay press and was used as an argument against vaccination. Figure 4 shows clearly that SARS-CoV-2 infection and – vaccination also has led to (a temporary) increase of interest in this topic.
Comment: Additionally the relevant publications that had addressed these questions experimentally are amiss. Just to provide some names of authors that had published ground-breaking data related T/B cell cross-reactivity and have to be mentioned in a related review:
Mateus/Grifoni/Weiskopf/Sette/Crotty; Swadling/Maini; Kundu/Lalvani; Braun/Loyal/Thiel; Song/Andrabi; Ng/Kassiotis; Niessl/Sekine/Buggert; LeBert/Bertoletti.
Response: We agree that in a review on the role of T and B cell immunity and cross protection from past and against future coronaviruses we have been incomplete in citing ground-breaking research from the research teams indicated by Reviewer #1. Please note that publications from the groups of Allesandro Sette (#75 in our original manuscript) and Nina Le Bert (#48) were included in the references. While in our Acknowledgement section we had indicated that “We apologize to all colleagues who have contributed immensely to our current understanding of the immunobiology of SARS-CoV-2 and other coronaviruses, but who’s work we could not include or cite due to space restrictions” (lines 483-485 original manuscript), this is not an excuse for this omission. We do agree that the research teams indicated by the reviewer should have been included explicitly in the review and in the References, which we have done in the revised manuscript (lines 174-191 of the clean revised manuscript).
Comment: Unfortunately, all questions addressed in this review are extensively and more qualitatively discussed elsewhere (especially by authors from the list above).
Response: When we were invited to contribute to the special issue of Microorganisms on “Coronaviruses: Past, Present, and Future” we wrote an Abstract on what would be the outline of our contribution and submitted that as a form of a pre-submission enquiry. When we got the OK from the guest-editor, we proceeded. We agree that in the review part of our manuscript there can and will be a certain overlap with previous reviews on this subject. With over 400,000 publications on COVID-19 this is rather unavoidable. Our views on the future of coronaviruses are our own. Others may have the same, or different views.
Comment: I recommend the authors to significantly overwork the manuscript and either change the focus or stratify the content.
Response: In our revised manuscript we have taken into account all comments made by Reviewer #1, as well as those of the other 3 reviewers. This has led to (further) stratification of the content.
Reviewer 2 Report
Comments and Suggestions for Authors
The approach to the previously quite described topic of the immune response to corona viruses in this manuscript is interestingly conceived and may represent importance in the future understandings. That is why the work must include all factors for prediction and not just some as the authors in this version have conceived.
Suggestions related to the manuscript concept:
1. As the authors repeatedly call for immune "protection", it is important to define what protection refers to: infection, illness, severe illness, transmission, etc. Only with a clear definition of this term will everything else be comparable.
2. Why is there no description-paragraph in the manuscript related to the influence of the evolutionary variability of one virus species, e.g. SARS-CoV-2? In lines 40-42, the authors mention the hypothesis that SARS-CoV-2 becomes a seasonal cold. Isn't this hypothesis quite dependent on the phylogenetic ie. bioinformatic predictions? In that case, we are talking about tens and hundreds of years.
3. Why do the authors consider all coronaviruses exclusively as a whole? Isn't the completely different tropism of these viruses important for the potential sustainability of immune memory?
4. If the hypothesis put forward by this manuscript is based on future protection, how has it not already happened through the 7 human coronaviruses that were introduced into the human population? Therefore, since there is not enough support for this hypothesis compared to previous experiences, then that part must be explained in more detail. On the other hand, do we have any proof of recombinations among the mentioned 7 types of coronaviruses?
5. Sections 2.1.2. and 2.1.3. describe therapy option for COVID-19. Here it is redundant and irrelevant because the work is conceived on prophylaxis.
6. Line 316. Therapy is again mentioned together with prophylaxis in the unclear aim of the author's statement and without reference
7. Section 3.1. is too long. I suggest shortening it.
8. In this manuscript, there should be a section describing the pancoronavirus vaccine.
Suggestions related to the validity of information:
1. Line 29: word „pneumonia“ is irrelevan
2. Lines 38-39: this is incorrect. WHO did not declare the end of the pandemic. Pay attention to the correct display of the announcement.
3. Section 2.2. with the Table 1 is incomplete. If the authors refer to some kind of systematic review, then it is necessary to fully describe the methodology with the avoidance of bias. Missing, for example: search keywords, search bases, methodology for removing duplicates, methods for excluding and including studies, etc.
4. Lines 439-440. A delicate and uncertain conclusion that has no clear reference support in this manuscript
Suggestions related to ambiguities:
1. Lines 40-48: the two scenarios are by no means related. so it is unclear what the authors wanted
2. Lines 126-127: where does this hypothesis come from? Reference
3. Conclusion is too long.
Minor points: Introduce abbreviations when they are first mentioned in the text
1. Line 109: What „S1“ abbreviation stands for?
2. Line 112: What „nAbs“ abbreviation stands for?
3. Line 208: publications instead of publication
Comments on the Quality of English LanguageMinor corrections
Author Response
Dear Reviewer
Thank you very much for taking the time and effort to review our manuscript. Please find the detailed point by point responses below. We have indicated the positions in the (clean version of the) manuscript where the changes were made. The corresponding revisions/corrections can also be found highlighted/in track changes in the re-submitted files.
Comment: The approach to the previously quite described topic of the immune response to corona viruses in this manuscript is interestingly conceived and may represent importance in the future understandings. That is why the work must include all factors for prediction and not just some as the authors in this version have conceived.
Response: It is correct that our review is restricted to an analysis of the immunological factors which may protect against future coronaviruses, or enhance disease severity. Other factors, such as virological, societal, environmental just to name a few, are important, maybe even more important than immune memory and cross-reactivity as such. This can be considered a limitation of our manuscript, and we have included such a statement in the Conclusions and outlook for the future section (lines 395-398 clean version of revised manuscript)
Suggestions related to the manuscript concept:
Comment 1: As the authors repeatedly call for immune "protection", it is important to define what protection refers to: infection, illness, severe illness, transmission, etc. Only with a clear definition of this term will everything else be comparable.
Response 1:We agree, and indeed have not clearly defined what we mean with immune protection. We define it as protection against (severe) COVID-19 caused by SARS-CoV-2 infection. In the revised manuscript we have made this explicit at all relevant places (initially in lines 11-13 of the clean version of the revised manuscript) .
Comment 2: Why is there no description-paragraph in the manuscript related to the influence of the evolutionary variability of one virus species, e.g. SARS-CoV-2? In lines 40-42, the authors mention the hypothesis that SARS-CoV-2 becomes a seasonal cold. Isn't this hypothesis quite dependent on the phylogenetic i.e. bioinformatic predictions? In that case, we are talking about tens and hundreds of years.
Response 2: We admit that a paragraph on the evolutionary variability of SARS-CoV-2 was missing. In the revised manuscript we have included such a paragraph, with a focus on RBD antibodies (lines 118-122 clean version of revised manuscript). (The lack of) evolutionary variability of T cell epitopes of the nucleocapsid protein is discussed in lines 213-214 and 219-223 of the clean revised manuscript.
Reviewer #2 is correct that in lines 40-42 we suggest that SARS-CoV-2 could become a common cold virus (“In an optimistic scenario it can be envisioned that within time SARS-CoV-2 can be added to the so-called cold cousins, i.e. the circulating coronaviruses which cause the common cold during winter seasons (229E, OC43, NL63, and HKU1)”).
The timeline of such an evolutionary conversion to a common cold coronavirus is difficult to predict and could take tens to hundreds of years. The appearance of the omicron variant, however, was a giant leap in the evolution of SARS-CoV-2 (Kandeel M, Mohamed MEM, Abd El-Lateef HM, Venugopala KN, El-Beltagi HS. Omicron variant genome evolution and phylogenetics. J Med Virol. 2022 Apr;94(4):1627-1632. doi: 10.1002/jmv.27515). Some virologists (as discussed in reference #4) are of the opinion that the omicron variant of SARS-CoV-2 can be considered a common cold virus.
We have added the following text to the revised manuscript: “It has to be emphasized that the timeline of the evolutionary conversion of a severe acute respiratory coronavirus to a common cold coronavirus is difficult to predict and potentially could take tens to hundreds of years” (lines 46-48 of the clean revised manuscript)
Comment 3: Why do the authors consider all coronaviruses exclusively as a whole? Isn't the completely different tropism of these viruses important for the potential sustainability of immune memory?
Response 3: Indeed, of the 7 hCoVs, only SARS-CoV-2 and SARS-CoV-1 use ACE-2 as their cellular receptor for viral entry, and the other 5 hCoVs use different cellular receptors. Even within SARS-CoV-2 and SARS-CoV-1 there is limited amino acid homology within the RBD. Outside of the RBD of the Spike protein, but especially in the Nucleocapsid protein, the T cell epitopes that have been identified have a high degree of homology across all 7 hCoVs. Therefore, we considered all hCoVs as a possible source of cross-reactivity. We have added this paragraph to the Concluding remarks and outlook for the future section of the revised manuscript (lines 389-394 of the clean revised manuscript).
Comment 4: If the hypothesis put forward by this manuscript is based on future protection, how has it not already happened through the 7 human coronaviruses that were introduced into the human population? Therefore, since there is not enough support for this hypothesis compared to previous experiences, then that part must be explained in more detail. On the other hand, do we have any proof of recombinations among the mentioned 7 types of coronaviruses?
Response 4: In section 4 of our manuscript (Concluding remarks and outlook for the future) we argue that there are indications that previous exposure to hCoVs did offer some kind of protection against SARS-CoV-2. We have provided these arguments in lines 376-382 of the clean version of the revised manuscript.
The second part of the question is very intriguing. There may be indications that the omicron strain of SARS-CoV-2 may have picked up sequences (by recombination or otherwise) from other coronaviruses. This is pure speculation, because these “other” coronaviruses, which may have been circulating in Africa for longer periods, have not been identified thus far, and may also not exist. We prefer not to include these speculations in our manuscript.
Comment 5: Sections 2.1.2. and 2.1.3. describe therapy option for COVID-19. Here it is redundant and irrelevant because the work is conceived on prophylaxis.
Response 5: IVIG and passive vaccination were included in the original manuscript because they serve as models to evaluate the relative contribution of (effector molecules of the) humoral immune system to protection against a viral infection. Because other reviewers had similar comments on inclusion of these sections, we have deleted them from the revised manuscript.
Comment 6: Line 316. Therapy is again mentioned together with prophylaxis in the unclear aim of the author's statement and without reference
Response 6: As indicated above, we have deleted sections 2.1.2 and 2.1.3 from the revised manuscript. Line 316 is the introduction for the section on antibody dependent enhancement (ADE). The antibodies that potentially could lead to enhancement of disease could have been induced by SARS-CoV-2 infection, by vaccination, or by passive administered antibodies (either prophylactic or therapeutic). We have revised the corresponding part of the manuscript (lines 279-284 of the clean revised manuscript)
Comment 7: Section 3.1. is too long. I suggest shortening it.
Response 7: As suggested, we have shortened section 3.1 considerably.
Comment 8: In this manuscript, there should be a section describing the pancoronavirus vaccine.
Response 8: We have included our views on a pancoronavirus vaccine in section 4. “Future coronavirus vaccines ideally would be pancoronavirus vaccines [74]. Such a vaccine cannot be solely based on Spike proteins or epitopes thereof because not all coronaviruses use ACE2 as cellular receptor. Even between SARS-CoV-2 and SARS-CoV-1 there is limited amino acid homology within the RBD. Outside of the RBD of the Spike protein, but especially in the Nucleocapsid protein, the T cell epitopes that have been identified have a high degree of homology across all 7 hCoVs. Those epitopes are included in a number of pancoronavirus vaccines currently under development [75]”. (lines 387-394 of the clean revised manuscript).
Suggestions related to the validity of information:
Comment 1: Line 29: word „pneumonia“ is irrelevant
Response 1: We agree and have deleted “pneumonia” from line 29.
Comment 2: Lines 38-39: this is incorrect. WHO did not declare the end of the pandemic. Pay attention to the correct display of the announcement.
Response 2: We admit that this wording is not correct. The WHO did declare that COVID-19 is now an established and ongoing health issue which no longer constitutes a public health emergency of international concern. We have adapted the text of the revised manuscript accordingly (lines 39-41 of the clean revised manuscript). That the term “public health emergency of international concern” as used by the WHO is under discussion (see e.g. Fan VY, Cash R, Bertozzi S, Pate M. The when is less important than the what: an epidemic scale as an alternative to the WHO's Public Health Emergency of International Concern. Lancet Glob Health. 2023 Oct;11(10):e1499-e1500. doi: 10.1016/S2214-109X(23)00314-5. Epub 2023 Aug 17. PMID: 37598696.) is not relevant in our context.
Comment 3: Section 2.2. with the Table 1 is incomplete. If the authors refer to some kind of systematic review, then it is necessary to fully describe the methodology with the avoidance of bias. Missing, for example: search keywords, search bases, methodology for removing duplicates, methods for excluding and including studies, etc.
Response 3: We agree that the data presented in Table 1 were not based upon a systematic review of the literature and that all factors which could have caused a bias were not taken into account. Reviewer #1 has indicated that Table 1 is redundant and should be removed. We therefore have deleted the complete Table 1 from the manuscript. We thank Reviewer #2 for the comments on Methodology.
Comment 4: Lines 439-440. A delicate and uncertain conclusion that has no clear reference support in this manuscript.
Response 4: The reviewer refers to our statement that “It is possible that existing memory T cells, generated during exposure to circulating hCoVs, including SARS-CoV-1 and MERS, have offered partial protection against SARS-CoV-2, which would explain the lower case fatality rate of SARS-CoV-2 as compared to SARS and MERS.” In the preceding sentences we write: “Our analysis of the literature shows and confirms that cytotoxic T cells against con-served epitopes on the SARS-CoV-2 spike protein and nucleocapsid protein are expanded during infection and do limit the severity of COVID-19 [14, 75]”. References #14 and #75 (numbering in original manuscript) are in support of our statement. In reference #14 for instance, Moss et al. write: “Thus, T cell epitopes are likely to be shared between viruses, and this cross-reactivity may be important in clinical protection”. The conserved T cell epitopes in the nucleocapsid protein are also shown in Table 3. We realize that a case fatality rate (of course) does not totally depend on the activity of cytotoxic T cells, and that many other factors, known and unknown, will contribute. That is why we start with “It is possible”. We believe that in this particular section of our manuscript (Concluding remarks and outlook for the future) this remark, (we admit is delicate) should be allowed.
In the revised manuscript we have rephrased this part so that the relation with the cited references becomes more clear. We also have softened the relation between T cell memory and case fatality rate: “Based on these findings, it is possible that existing memory T cells, generated during exposure to circulating hCoVs, including SARS-CoV-1 and MERS, have offered partial protection against SARS-CoV-2, and thus could have contributed to a lower case fatality rate of SARS-CoV-2 as compared to SARS and MERS”(lines 378-382 of the clean version of the revised manuscript)
Suggestions related to ambiguities:
Comment 1: Lines 40-48: the two scenarios are by no means related. so it is unclear what the authors wanted.
Response 1: We understand that it is unclear what was meant by the two different scenarios. We have re-written them, so that the relation is clear now (lines 42-51 of the clean version of the revised manuscript).
Comment 2: Lines 126-127: where does this hypothesis come from? Reference.
Response 2: Lines 124-127 of the original manuscript read: “The immune response of agammaglobulinemia patients during SARS-CoV-2 infection and the outcome of COVID-19 could serve as a model to further investigate the role of humoral immunity and thus help predict future outcomes”. This line serves as an introduction for the following paragraph. Patients with XLA lack B cells and thus are unable to generate a humoral immune response. Their host defence to infection with SARS-CoV-2 therefore totally depends on T cell immunity. Because our original wording was unclear we have rephrased the relevant section (lines 125-131 of the clean revised manuscript).
Comment 3: Conclusion is too long.
Response 3: We have tried to keep the Concluding remarks and outlook for the future section as compact as possible. In the revised version of the manuscript we had to include the response to several comments made by the different Reviewers, so this remains a significant part of the manuscript.
Minor points: Introduce abbreviations when they are first mentioned in the text
Comment 1: Line 109: What „S1“ abbreviation stands for?
Response 1: We have explained the S1 abbreviation when first used by revising the lines 107-108 of the clean version of the revised manuscript as follows: “Furthermore, compared to mild cases, severe cases had significantly higher levels of IgG antibodies directed against the S1 subunit of the spike protein (S1)”.
Comment 2: Line 112: What „nAbs“ abbreviation stands for?
Response 2: nAbs stands for neutralizing antibodies, which was explained in line 106.
Comment 3: Line 208: publications instead of publication
Response 3: The word “publication” in the legend of Table 1 indeed should have been “publications”. Because of the remarks of other reviewers on Table 1, this Table was deleted.
We thank the reviewer for the critical evaluation of our manuscript, for the constructive criticism and the useful suggestions and comments. This has allowed us to revise and improve our manuscript.
Reviewer 3 Report
Comments and Suggestions for Authors
The authors present a review on immune dynamics of coronaviruses with future implications. The subject matter fits well with the focus of the special issue. The manuscript is very well written and presents an interesting aggregation of information. I only have some minor points for the authors to consider in improving their manuscript.
Line 139
This is the first time the abbreviation IVIG appears and should be defined before use in the text.
Line 318
Corona virus has been separated which is not consistent with the rest of the manuscript.
Line 358-368
The description of the ADE in flaviruses here does not seem so have a direct bearing unless hypothesised that the mechanism is similar in coronaviruses. In any case the linkage or purpose of this is not quite clear and could be improved.
Comments on the Quality of English LanguageEnglish language is fine
Author Response
Dear Reviewer
Thank you very much for taking the time and effort to review our manuscript. Please find the detailed point by point responses below. We have indicated the positions in the (clean version of the) manuscript where the changes were made. The corresponding revisions/corrections can also be found highlighted/in track changes in the re-submitted files.
Comment: The authors present a review on immune dynamics of coronaviruses with future implications. The subject matter fits well with the focus of the special issue. The manuscript is very well written and presents an interesting aggregation of information. I only have some minor points for the authors to consider in improving their manuscript.
Response: We thank the reviewer for these comments.
Comment: Line 139; This is the first time the abbreviation IVIG appears and should be defined before use in the text.
Response: That was an omission. We have defined IVIG (intravenous immunoglobulins) in the revised text (line 143 of the clean revised manuscript).
Comment: Line 318; Corona virus has been separated which is not consistent with the rest of the manuscript.
Response: Our apologies! We had tried so hard to be consistent, but this one slipped through. Thank you for noting and we have corrected.
Comment: Line 358-368; The description of the ADE in flaviruses here does not seem so have a direct bearing unless hypothesized that the mechanism is similar in coronaviruses. In any case the linkage or purpose of this is not quite clear and could be improved.
Response: We have rephrased the introduction of this section, now more clearly stating what is observed in case of dengue virus. We do indicate that the mechanism is similar in coronaviruses (lines 318-330 in clean revised manuscript).
We thank the reviewer for the critical evaluation of our manuscript, for the constructive criticism and the useful suggestions and comments. This has allowed us to revise and improve our manuscript.
Reviewer 4 Report
Comments and Suggestions for Authors
This article tried to provide a comprehensive review as to immunological studies of coronaviruses infection. The article wants to reply which of humoral and T-cell immunity would be more important for providing protection against coronaviruses, especially to SARS-CoV-2. Immunological phenomena of ADE and the original antigen sin were also discussed to improve our administration for prevention and treatment of coronaviruses. The contents should be interesting for coworkers and some innovative ideas were included in. Minor revision comments are listed below.
Line 9, “The alpha coronaviruses NL63 and 229E and the beta coronaviruses OC34 and HKU1” is not a complete sentence.
Line 30, “millions of people” is incorrect. Billions of people worldwide should have been affected in the pandemic.
Line 35, “Thanks mainly to the development and implementation of effective vaccines, the pandemic could be brought under control”. There should be some controversial in this statement. I believe the real brake for the pandemic should be the emergence of Omicron strain, which infected people but was not lethal. The Director of WHO described the Omicron as 2012 Christmas gift to the people of the whole world.
Line 51, “The subfamily of Coronavirinae further can be divided into three major groups” is not correct. According to Coronaviridae Study Group of the International Committee on Taxonomy of Viruses, there were four species of coronaviruses, including alpha, beta, gamma and delta. SARS-CoV-2 belongs the species Severe acute respiratory syndrome-related coronavirus ( Nat Microbiol. 2020 Apr;5(4):536-544. doi: 10.1038/s41564-020-0695-z. Epub 2020 Mar 2. PMID: 32123347; PMCID: PMC7095448).
Line 357: In the section 3.1.2 contents do not meet the title. This section mainly discussed mechanism of ADE and glycosylation of antibodies, no much In vivo factors were mentioned. What the author wanted to discuss in the section with the title ”In vivo indications for antibody dependent enhancement“?
Line 417-421: This section needs rearrangement of sentence structure and clearer narration.
Comments on the Quality of English LanguageA minor language revision is needed.
Author Response
Dear Reviewer
Thank you very much for taking the time and effort to review our manuscript. Please find the detailed point by point responses below. We have indicated the positions in the (clean version of the) manuscript where the changes were made. The corresponding revisions/corrections can also be found highlighted/in track changes in the re-submitted files.
Comment: This article tried to provide a comprehensive review as to immunological studies of coronaviruses infection. The article wants to reply which of humoral and T-cell immunity would be more important for providing protection against coronaviruses, especially to SARS-CoV-2. Immunological phenomena of ADE and the original antigen sin were also discussed to improve our administration for prevention and treatment of coronaviruses. The contents should be interesting for coworkers and some innovative ideas were included in. Minor revision comments are listed below.
Response: We thank the Reviewer for these comments.
Comment: Line 9, “The alpha coronaviruses NL63 and 229E and the beta coronaviruses OC43 and HKU1” is not a complete sentence.
Response: This was a mistake! We have corrected the relevant part of the Abstract (lines 7-9 of the clean revised version of the manuscript).
Comment: Line 30, “millions of people” is incorrect. Billions of people worldwide should have been affected in the pandemic.
Response: We agree. We did not want to downplay the pandemic, and indeed billions of people were affected, if not by the disease itself then certainly by the social and economic impact.
Comment: Line 35, “Thanks mainly to the development and implementation of effective vaccines, the pandemic could be brought under control”. There should be some controversial in this statement. I believe the real brake for the pandemic should be the emergence of Omicron strain, which infected people but was not lethal. The Director of WHO described the Omicron as 2012 Christmas gift to the people of the whole world.
Response: We couldn’t find a quote of the director of the WHO saying or implying that the omicron variant could be considered a Christmas gift. The German epidemiologist Karl Lauterbach, in a Tweet, described the Covid-19 strain “Omicron” as an “early Christmas gift”, saying it could mean an end to the pandemic.
(Pdf with Tweet added)
Based on the above, we have used neutral wording when describing how the pandemic was brought under control: “Thanks mainly to the development and implementation of effective vaccines, and probably also the emergence of the omicron variant, the pandemic could be brought under control”. (lines 35-36 clean version of the revised manuscript).
Comment: Line 51, “The subfamily of Coronavirinae further can be divided into three major groups” is not correct. According to Coronaviridae Study Group of the International Committee on Taxonomy of Viruses, there were four species of coronaviruses, including alpha, beta, gamma and delta. SARS-CoV-2 belongs the species Severe acute respiratory syndrome-related coronavirus ( Nat Microbiol. 2020 Apr;5(4):536-544. doi: 10.1038/s41564-020-0695-z. Epub 2020 Mar 2. PMID: 32123347; PMCID: PMC7095448).
Response: We agree. We should have referred to the International Committee on Taxonomy of Viruses and have revised the text as well as the reference accordingly.
Comment: Line 357: In the section 3.1.2 contents do not meet the title. This section mainly discussed mechanism of ADE and glycosylation of antibodies, no much In vivo factors were mentioned. What the author wanted to discuss in the section with the title ”In vivo indications for antibody dependent enhancement“?
Response: Also based on comments made by other reviewers, we have rephrased this section, and also refer to the in vivo experiments with feline coronavirus vaccination (lines 318-330 of the clean revised manuscript).
Comment: Line 417-421: This section needs rearrangement of sentence structure and clearer narration.
Response: We agree that the narrative of this section is rather poor. We have rephrased this part, and shortened it (lines 355-362 clean version of the revised manuscript).
Comment: Comments on the Quality of English Language: A minor language revision is needed.
Response: We have asked a colleague microbiologist, a native English speaker, for language revision and have indicated her assistance in the Acknowledgements.
We thank the reviewer for the critical evaluation of our manuscript, for the constructive criticism and the useful suggestions and comments. This has allowed us to revise and improve our manuscript.

Round 2
Reviewer 1 Report
Comments and Suggestions for Authors
Line 136: neutralizing antibodies are not only RBD specific. What about cross-neutralizing/cross-reactive antibodies?
Line 252f and 273f: Tarke several hundred epitopes versus Ferretti 3 epitopes - these parts are lacking a proper differentiation of CD4 and CD8 T cells as well as the fact, that SARS-CoV-2 specific CD8 T cells are less detectable and relevant in blood but rather in tissues. See also the cited Swadling paper as well as the according Bertoletti paper.
Line 348f: The authors describe ADE but I miss a link it to any findings related to SARS-CoV-2.
The authors should further stratify the manuscript towards answering the questions addressed in the abstract.
Author Response
Dear Reviewer
First and foremost we would like to express our gratitude for taking the time to review our submission. Thanks to your insightful comments and feedback we have been able to refine our work.
Comment 1: Line 136: neutralizing antibodies are not only RBD specific. What about cross-neutralizing/cross-reactive antibodies?
Reply 1: Regarding the specificity of neutralising antibodies we appreciate your thoughtful feedback. We are grateful fort his comment, because it made us check a relevant paper by Chen et al. (Chen Y, Zhao X, Zhou H, Zhu H, Jiang S, Wang P. Broadly neutralizing antibodies to SARS-CoV-2 and other human coronaviruses. Nat Rev Immunol. 2023 Mar;23(3):189-199. doi: 10.1038/s41577-022-00784-3.).
We have revised our manuscript as follows:
Neutralizing antibodies mostly are directed against the receptor binding domain (RBD) of the SARS-CoV-2 spike protein (indicated in Figure 2), but can also be directed against the N-terminal domain, or the stem helix region or the fusion peptide region in the S2 subunit of the spike protein [21]. Lines 126-129 of the clean revised manuscript.
By adding this additional specificities, we believe the spectrum of neutralizing antibodies now has been described fully.
Comment 2: Line 252f and 273f: Tarke several hundred epitopes versus Ferretti 3 epitopes - these parts are lacking a proper differentiation of CD4 and CD8 T cells as well as the fact, that SARS-CoV-2 specific CD8 T cells are less detectable and relevant in blood but rather in tissues. See also the cited Swadling paper as well as the according Bertoletti paper.
Reply 2: We appreciate your remarks on the discrepancy between the number of T cell epitopes found by Tarke et al. and Ferrett. Also the comments on CD4 and CD8 epitopes, and tissue location as discussed by Swadling and Bertoletti are well taken and is gratefully acknowledged. We have now have elaborated on these aspects and revised the manuscript as follows:
Tarke et al., by using overlapping peptide spanning all structural and non-structural viral proteins, identified several hundred T cell epitopes for CD4+ T cells and CD8+ T cells, highlighting the diverse T cell response to the virus [35]. Additionally, abortive infections can exist where T cell responses may clear the virus before sufficient viral replication or antibody production have taken place. This process is explored in a study by Swadling et al. who found that the T cells involved in this process are mainly directed against non-structural proteins of the replication-transcription complex [36]. It should be kept in mind that most studies on T cell epitope mapping are performed with peripheral blood T cells. The T cells which have infiltrated infected tissue such as the lung may have a different specificity pattern [37]. Lines 198-207 of the clean revised manuscript.
In the subsequent part of the manuscript where the data from Feretti et al. are discussed we have revised the opening sentence as follows:
As indicated above, the Spike protein contains many T cell epitopes, of which 3 can be considered major epitopes as defined by Ferretti et al., termed YLQ, KCY and QYI [38]. Lines 223-224 of the clean revised manuscript.
Moreover, in a later section (lines 276-277) of the revised manuscript) we also have referred to this discussion
As indicated above, apart from the Spike and N protein, many T cell epitopes in SARS-CoV-2 are also found in non-structural parts (NSP) like ORF1ab or ORF3a [35, 52].
Comment 3: Line 348f: The authors describe ADE but I miss a link it to any findings related to SARS-CoV-2.
Reply 3: We agree that we hadn’t made the direct link to SARS-CoV-2. In the revised manuscript we have added a paragraph summarizing the findings on the outcome of a SARS-CoV-2 infection in vaccinated versus non-vaccinated patients, as well as the relevant references.
When ADE would occur in case of SARS-CoV-2 infection, it would be expected that after vaccination COVID-19 would take a more severe course. However, studies on break-through infections [60, 61] as well as case control studies [62] have shown that vaccinated individuals with breakthrough COVID-19 infections experience milder symptoms, with fewer hospital and ICU admission and lower mortality rates as compared to unvaccinated individuals [63]. Lines 357-362 of the clean revised manuscript.
Comment 4: The authors should further stratify the manuscript towards answering the questions addressed in the abstract.
Reply 4: We appreciate your thoughtful feedback on how to better stratify our manuscript to better address the issues raised in the abstract. With the implementation of the suggestions revisions we believe our manuscript now addresses all issues from the abstract. We have improved the coherence and depth of our manuscript by clarifying some points, and strengthening the conclusions in accordance with your suggestions.